# Evaluation of the Pathogenic Potential of *Escherichia coli* Strains Isolated from Eye Infections

**DOI:** 10.3390/microorganisms10061084

**Published:** 2022-05-25

**Authors:** Pedro Henrique Soares Nunes, Tiago Barcelos Valiatti, Ana Carolina de Mello Santos, Júllia Assis da Silva Nascimento, José Francisco Santos-Neto, Talita Trevizani Rocchetti, Maria Cecilia Zorat Yu, Ana Luisa Hofling-Lima, Tânia Aparecida Tardelli Gomes

**Affiliations:** 1Laboratório Experimental de Patogenicidade de Enterobactérias (LEPE), Disciplina de Microbiologia, Departamento de Microbiologia, Imunologia e Parasitologia (DMIP), Escola Paulista de Medicina (EPM), Universidade Federal de São Paulo (UNIFESP), Sao Paulo 04023-062, Brazil; phs.nunes@unifesp.br (P.H.S.N.); tiago.valiatti@unifesp.br (T.B.V.); carolina.mello@unifesp.br (A.C.d.M.S.); jullia.nascimento@unifesp.br (J.A.d.S.N.); jfs.neto@unifesp.br (J.F.S.-N.); 2Laboratório de Oftalmologia (LOFT), Departamento de Oftalmologia e Ciências Visuais, Escola Paulista de Medicina (EPM), Universidade Federal de São Paulo (UNIFESP), Sao Paulo 04023-062, Brazil; talitaunesp@gmail.com (T.T.R.); loft.unifesp@gmail.com (M.C.Z.Y.); analhofling@gmail.com (A.L.H.-L.); 3Laboratório Alerta, Disciplina de Infectologia, Departamento de Medicina, Escola Paulista de Medicina (EPM), Universidade Federal de São Paulo (UNIFESP), Sao Paulo 04039-032, Brazil

**Keywords:** ExPEC, bacterial pathogenicity, eye infection, virulence genes, antibiotic resistance

## Abstract

While primarily Gram-positive bacteria cause bacterial eye infections, several Gram-negative species also pose eye health risks. Currently, few studies have tried to understand the pathogenic mechanisms involved in *E. coli* eye infections. Therefore, this study aimed to establish the pathogenic potential of *E. coli* strains isolated from eye infections. Twenty-two strains isolated between 2005 and 2019 from patients with keratitis or conjunctivitis were included and submitted to traditional polymerase chain reactions (PCR) to define their virulence profile, phylogeny, clonal relationship, and sequence type (ST). Phenotypic assays were employed to determine hemolytic activity, antimicrobial susceptibility, and adhesion to human primary corneal epithelial cells (PCS-700-010). The phylogenetic results indicated that groups B2 and ST131 were the most frequent. Twenty-five virulence genes were found among our strains, with *ecp*, *sitA*, *fimA*, and *fyuA* being the most prevalent. Two strains presented a hemolytic phenotype, and resistance to ciprofloxacin and ertapenem was found in six strains and one strain, respectively. Regarding adherence, all but one strains adhered in vitro to corneal cells. Our results indicate significant genetic and virulence variation among ocular strains and point to an ocular pathogenic potential related to multiple virulence mechanisms.

## 1. Introduction

Bacteria are among the most relevant pathogens in ophthalmology, being responsible for numerous eye diseases, such as blepharitis, conjunctivitis, keratitis, and endophthalmitis. It is known that a healthy ocular surface contains a paucibacterial microbiota [1,2] and that some factors such as the misuse of contact lenses, bad hygiene, eye injuries, and a compromised immunologic system may favor the growth of potentially pathogenic species [3]. While Gram-positive species are more frequently identified as causative agents in eye infections, Gram-negative species also pose a threat to ocular health [1,2,3].

*Escherichia coli*, a member of the Enterobacteriaceae family and the Enterobacterales order, is a Gram-negative bacillus found mainly in the human gut microbiota, but that may also be found in other sites such as the human eye. However, its presence on this site remains not quite well explained [1]. Strains of *E. coli* can be classified as commensal or pathogenic, with commensal strains establishing a symbiotic relationship with the human host. In contrast, pathogenic strains may carry many virulence markers that, in turn, enable their capacity to cause diseases in intestinal or extraintestinal sites [4,5].

Pathogenic *E. coli* strains can be classified depending on the infection site and divided into pathotypes, primarily based on the virulence markers they might carry but also on phenotypic characteristics. Strains capable of causing intestinal infections are classified as intestinal pathogenic *E. coli* (InPEC) and are further divided into the following pathotypes: typical and atypical enteropathogenic *E. coli* (EPEC), Shiga toxin-producing *E. coli* (STEC), enterotoxigenic *E. coli* (ETEC), typical and atypical enteroaggregative *E. coli* (EAEC), enteroinvasive *E. coli* (EIEC), diffusely adherent *E. coli* (DAEC), and adherent-invasive *E. coli* (AIEC) [6,7,8]. Strains related to extraintestinal infections can be classified according to their site of isolation, namely uropathogenic *E. coli* (UPEC), neonatal meningitis-associated *E. coli* (NMEC), and septicemic *E. coli* (SEPEC) [9].

Extraintestinal pathogenic *Escherichia coli* (ExPEC) strains can carry a plethora of virulence genes that enable their survival on extraintestinal sites. Based on animal models of virulence, Johnson et al. [10] proposed that ExPEC strains with full virulence potential, that is, capable of infecting immunocompetent individuals, should simultaneously carry at least two out of five previously specific virulence markers—namely, *pap*, *sfa*, *afa/dra*, *iuc/iut*, and *kpsMTII*. Furthermore, Spurbeck et al. [11] proposed that the simultaneous presence of the genes *chuA, fyuA, vat,* and *yfcV* are epidemiologically related to uropathogenicity. It is important to notice that, despite these proposals, *E. coli* strains devoid of all these virulence markers may also cause extraintestinal infections [12].

In recent times, an interest in “hybrid” *E. coli* strains has emerged. A hybrid *E. coli* strain is defined by the presence in a single strain of a combination of virulence markers previously believed to be found only in specific intestinal or extraintestinal *E. coli* pathotypes [13]. Besides likely resulting in increased pathogenic potential, the hybrid genotype may also confer the ability to a single strain to infect and cause intestinal and extraintestinal diseases [14,15,16], thus reinforcing the importance of surveillance among pathogenic *E. coli* strains. However, due to their recent discovery, the frequency of hybrid strains in a clinical setting is yet to be defined.

*E. coli* can be considered a relevant ocular pathogen, but the origins of ocular pathogenic *E. coli* and the mechanisms involved in the ocular pathogenicity of ExPECs are not quite well defined, mainly due to the lack of published works on this theme. Furthermore, the genetics of ocular *E. coli* strains are largely understudied, and more studies are required to understand better the genetic evolution of these strains. Therefore, our main goal was to analyze the genotypic and phenotypic characteristics of *E. coli* strains isolated from eye infections.

## 2. Materials and Methods

### 2.1. Ethics

This retrospective study was approved by the Ethics Committee of the Federal University of São Paulo/UNIFESP, São Paulo, Brazil (CEP 7829241219 (31 December 2019)). For this study, formal consent was not required as all strains used were collected during a clinical routine at the Hospital São Paulo and kept in a microbiological bank at the Ophthalmology Laboratory (LOFT) after routine laboratory procedures.

### 2.2. Bacterial Strains

Twenty-two strains were isolated from April 2005 to June 2019 from patients diagnosed with keratitis or conjunctivitis at Hospital São Paulo (HSP), UNIFESP. Five strains were isolated from corneal scrapings obtained from five patients with infectious keratitis, and sixteen were isolated from conjunctival swabs from fourteen patients with infectious conjunctivitis. One strain was isolated from the contact lenses of a patient with keratitis. All collected strains were stored at −80 °C for research purposes. Strains used in this study were labeled following the resulting disease—K for keratitis and C for conjunctivitis. Strains C-3a and C-3b were obtained from a single patient, but both strains were included in this study, based on their molecular and morphological divergences. All strains were cultivated in lysogeny broth (LB) and MacConkey agar to verify their purity. Species-level identification was confirmed using the Matrix-Assisted Laser Desorption Ionization—Time of Flight—Mass Spectrometry (MALDI-TOF MS) technique, performed in the Microflex LT mass spectrometer (Bruker Daltonics, Billerica, MA, USA). Clinical details from patients are presented in Appendix A.

### 2.3. Phylogenetic Analysis, Clonal Relationship, and Sequence Typing

The multiplex polymerase chain reaction (PCR) method proposed by Clermont et al. [17] was employed to determine to which *E. coli* phylogenetic group each strain belonged. Briefly, four genes—namely *chuA, yjaA, arpA,* and TspE4.C2—were used in a quadruplex reaction. All PCRs in this study were prepared using GoTaq^®^ Green Master Mix (Promega, Madison, WI, USA) with 10 pmol of primers and 1 µL of DNA template obtained by bacterial thermal lysis in sterilized water for PCR. The resulting products were analyzed through agarose gel electrophoresis (1%) in 1× Tris-Borate-EDTA (TBE) buffer, stained with ethidium bromide (10 µg/mL) for 15 min, and photo-documented in the Gel Doc XR+ Gel Documentation System (Bio-Rad, Hercules, CA, USA).

The clonal relationship of the strains was determined by Random Amplified Polymorphic DNA (RAPD). To this end, two PCRs were performed, each with a single primer, 1254 (5′-CCGCAGCCAA-3′) and 1283 (5′-GCGATCCCCA-3′). Primers were used based on previous works by Pacheco et al. [18] and Nilsen et al. [19], respectively.

The multiplex PCR proposed by Doumith et al. [20] was used to determine the Sequence Type (ST), which included the STs 131, 69, 73, and 95.

### 2.4. Virulence Genes Profiles

The virulence genes profiles of the strains were determined by PCR for a total of 42 virulence-associated genes, which were selected based primarily on their association with the virulence of ExPEC strains. The genes used as EPEC, EAEC, ETEC, EIEC, and STEC diagnostic markers were also included to search for possible hybrid strains. The genes were included based on Nascimento et al. [21]. All PCRs were performed following the previously described methodology.

The following genes were included in this study: *afaBCIII, afaE-VIII, aggR, bfpB, bmaE, cf29A, chuA, cnf1, cvaC, eae, ecp, ehxA, eltA, escV, estA, fimA, fyuA, hlyA, hra, ibeA, iha, invE, ireA, iroN, irp2, iucD, kpsMTII, kpsMTIII, ompA, ompT, papA, papC, pic, sat, sfa DE, sitA, stx1, stx2, traT, tsh, vat,* and *yfcV*.

### 2.5. Hemolytic Activity

The protocol established by Beutin et al. [22] was employed to evaluate the hemolytic activity of the strains. Briefly, the strains were first cultivated in 3 mL of Tryptic Soy Broth (TSB) at 37 °C for 18 h and subsequently inoculated on a single blood agar plate, prepared with Tryptic Soy Agar supplemented with washed sheep red blood cells (Laborclin, Brazil) (final concentration of 5%), and 10 mM CaCl_2_. The inoculated plate was analyzed after 3 h and 24 h of incubation at 37 °C. Strains EDL933 and CFT073 were used as positive controls for enterohemolysin and α-hemolysin, respectively, and the *Klebsiella* spp. strain K33 was used as a negative control.

### 2.6. Antibiotic Susceptibility

The Kirby-Bauer disc-diffusion method [23] was used to determine the susceptibility of all 22 strains. To do so, strains were cultivated on MacConkey agar for 24 h, at 37 °C, and their resulting colonies were diluted in saline solution (0.85%) until a 0.5 McFarland standard turbidity was attained. The resulting bacterial suspensions were inoculated onto Mueller Hinton agar plates and discs were applied with sterile forceps. Plates were then incubated for 18 h, at 37 °C. Results were interpreted according to the criteria established by the Brazilian Committee on Antimicrobial Susceptibility Testing (BrCAST/EUCAST). The breakpoints used were those defined by the European Committee on Antimicrobial Susceptibility Testing (EUCAST) (https://www.eucast.org/ (accessed on 13 November 2021)). The following antimicrobials were tested: amikacin, aztreonam, cefepime, cefoxitin, cefotaxime, ceftazidime, ciprofloxacin, gentamicin, imipenem, ertapenem, meropenem, and tigecycline.

### 2.7. Cell Culture and In-Vitro Adherence to Human Corneal Epithelial Cells Assay

Qualitative and quantitative assays with normal human primary corneal epithelial cells (HCECs) PCS-700-010, obtained from ATCC (ATCC, Manassas, VA, USA), were performed to evaluate the ability of *E. coli* strains to colonize HCECs. Primary HCECs were maintained in Dulbecco’s Modified Eagle Medium Nutrient Mixture F-12 (DMEM/F12) with GlutaMAX^TM^-I (Gibco, Grand Island, NY, USA) supplemented with Corneal Epithelial Cell Growth Kit (ATCC), 10% bovine fetal serum (BFS) (Gibco, Brazil), and 1× PSN antibiotic mixture (penicillin—5 mg/mL, streptomycin—5 mg/mL, and neomycin—10 mg/mL (Gibco, Carlsbad, CA, USA)). Cells were cultivated and maintained in 75 mL cell culture flasks at 37 °C in an atmosphere of 5% CO_2_.

For quantitative analysis, assays were performed following the protocol used by Valiatti et al. [24] with some modifications. Twenty-four-well microplates were prepared with ~10^5^ cells/per well, and, after 72 h of incubation, the medium contained in each well was discarded and cells were washed three times with sterile phosphate-buffered saline (PBS). Then, 1 mL of DMEM/F12 with Corneal Epithelial Cell Growth Kit and 2% BFS was added to each well and subsequently inoculated with 20 µL of each bacterial suspension containing approximately 10^8^ colony-forming unities per milliliter (CFU/mL). After 3 h incubation at 37 °C, the growth medium was discarded, and each well was washed three times with sterile PBS. Next, cell lysis was obtained by adding 1 mL of sterile double-distilled water to each well and incubating for 30 min at 37 °C. The wells’ contents were collected, serial diluted, and plated onto MacConkey agar for bacterial counting.

For qualitative assays, glass coverslips were added to each well of 24-well plates and all previously mentioned steps were repeated, except for the cell lysis process. Instead, cells were fixed with methanol at room temperature, stained with May-Grünwald and Giemsa (Merck, Darmstadt, Germany), and visualized under immersion oil light microscopy. All assays were performed in technical and biological replicates.

Strain CFT073 was used as an ExPEC positive control, and an uninfected well was considered a negative control.

## 3. Results

### 3.1. Phylogenetic Analysis, Clonal Relationship, and Sequence Typing

In total, thirteen (59%) strains were identified as belonging to the phylogenetic group B2, three to group A, two to group D, two to group F, one to group C, and one to group B1. None of the strains belonged to group E. Regarding Sequence Typing, eleven (50%) strains were identified as ST131, while the remaining strains were negative for all STs included in the quadruplex (Table 1).

Based on the sequence typing results, all eleven ST131 strains were submitted to RAPD typing. Based on the resulting profiles, two genetic clusters were identified. Strains C-3a, C-3b, C-4, and C-11 shared the same profiles with both primers, thus constituting a cluster. Strains C-8, C-9, and K-5 shared the same profiles with both primers and formed a second cluster. In addition, both clusters appeared to be genetically related to some degree since they shared an identical amplification pattern in one of the PCRs. The other strains presented distinct profiles.

By comparing phylogroup and disease, we could observe that groups B2, D, and F made up 87.5% of all conjunctivitis strains and three out of six keratitis strains (50%) were from group B2.

### 3.2. Virulence Genes Profile

Twenty-five (62.5%) out of the 42 genes screened were identified in at least one strain of our collection. The *ecp, fyuA, sitA, fimA, ompA,* and *irp2* genes were the most prevalent among our strains, being present in 100% (*ecp*), 90.9% (*sitA*), 86.3% (*fyuA, fimA*), and 81.8% (*irp2, ompA*) of the strains. The *iha, iucD, chuA, ompT, sat, yfcV, kpsMTII, iroN,* and *traT* genes were also identified in a significant percentage of strains, with identification ranging from 77.2% to 50% (Figure 1). We did not identify genes associated with the InPEC pathotypes, such as *eae, aggR, stx1, stx2, eltA, estA, invE,* and *bfpB,* among our strains, thus indicating the absence of hybrid strains in our collection. Regarding the individual virulence profile of each strain, we observed that most strains—17 out of 22—carried at least 12 virulence genes. Strain C-4 carried the highest number of genes, with 18 genes.

Seventeen strains presented at least two of the five intrinsic virulence markers proposed by Johnson et al. [10] to define ExPEC intrinsic virulence. Of the remaining five strains, four carried one of these markers, while strain K-2 was devoid of all five markers. Furthermore, based on Spurbeck et al. [11] proposal regarding uropathogenicity, we also screened our strains for the four virulence markers related to more efficient urinary tract colonization. In total, 12 strains were classified as potentially uropathogenic as they carried, simultaneously, at least three of the four proposed virulence markers. From these 12 strains, 10 also fell under the ExPEC intrinsic virulence classification. Moreover, K-3 and K-4 were the only strains to carry all four UPEC virulence markers (Table 2).

### 3.3. Hemolytic Activity

Only two strains presented hemolytic activity, namely strains K-3 and C-15. Although both strains formed a clear halo of hemolysis around the colonies, the halo developed by the former strain was observed after 3 h of incubation, while the halo produced by the latter strain was detected only after 24 h (Figure 2). Interestingly, both strains lacked the *hlyA* and *ehxA* genes. Despite being positive for *hlyA*—a gene associated with the α-hemolysis phenotype—strains C-3a, C-3b, and C-4 were non-hemolytic.

### 3.4. Antibiotic Susceptibility

Fifteen (68.2%) of the strains included in the study were sensitive to all the antimicrobials tested; six (27.3%) were resistant to ciprofloxacin, one (4.5%) was resistant to ertapenem, and one (4.5%), which was resistant to ciprofloxacin, was also resistant to gentamicin (Appendix A).

### 3.5. In-Vitro Adherence to Human Corneal Epithelial Cells

Except for C-14, all strains were able to adhere—with variable intensities—to human corneal epithelial cells. The mean interaction ranged from 10^5^ CFU/mL to 10^7^ CFU/mL. Among the strains, C-5 and C-15 were the most adherent, while strains C-1 and C-2 were the least adherent (Figure 3B).

Under oil immersion light microscopy, we could observe that all strains interacted with HCECs in vitro, with clearly definable differences regarding adherence level. Some strains visibly demonstrated a lower level of adherence, with only a few visible bacteria adhering to HCECs, while others presented a higher level of adherence, with bacteria distributed diffusely on cells. In addition, except for strain C-6, no specific adherence pattern was identified among the strains (Figure 3A). Curiously, strain C-6 presented a pattern that resembled the aggregative adherence pattern of EAEC strains. Following this phenotypical presentation and the virulence profile results, we also screened this strain for genes associated with atypical EAEC strains—namely *aatA*, *aaiA*, *aaiC*, and *aaiG*—but none of these genes were identified in the strain (data not shown). None of the tested strains promoted apparent morphological changes in the infected cells.

## 4. Discussion

The *E. coli* strains associated with eye infections are understudied, despite their continuous—yet not so frequent—isolation from this site [25,26,27]. To enhance the knowledge regarding this topic, in the present work, we performed a molecular and phenotypic evaluation of diverse virulence aspects in a collection of *E. coli* strains isolated from cases of keratitis and conjunctivitis.

It is a consensus that *E. coli* strains isolated from extraintestinal infections share diverse traits related to their capacity to cause infections out of the intestinal site. In this sense, phylogroup B2 is pointed out as the most frequent group associated with extraintestinal infections due to the higher prevalence of specific virulence factors related to the extraintestinal pathogenicity that these strains commonly bear [28,29].

Our phylogenetic findings are similar to those presented by several authors, in which strains of group B2, group D, and group F are more commonly associated with extraintestinal infections [8,30,31,32]. It has been proposed that group B2 is more prevalent in extraintestinal infections due to a higher prevalence of virulence genes related to extraintestinal pathogenicity [29,30,33]. This observation also correlates with our findings, given that all strains with the highest number of ExPEC virulence markers were identified as belonging to group B2. Many studies have also observed that UPEC strains are more frequently identified as belonging to group B2 than to other phylogenetic groups [34,35]. Our results align with these observations as all strains classified as potentially uropathogenic in our virulence profile analysis were from group B2. It is worth noting that strains of groups A and C also carried ExPEC virulence markers, but to a lesser extent than the previously mentioned groups. This is especially noteworthy given that, in recent years, it has been observed a more expressive presence of pathogenic strains of group A—a group primarily associated with commensal strains—in reports of extraintestinal infections caused by *E. coli*. Such an increase in infections caused by strains of group A has been reported not only by our group but also by other authors worldwide [14,21,36,37].

ST131 is currently the most well-established Sequence Type among *E. coli* strains that cause extraintestinal infections globally [38,39,40]. Furthermore, in recent years, *E. coli* ST131 has been considered a high-risk clone linked to increased resistance rates to fluoroquinolones and β-lactams [41,42,43]. This is also reinforced by our results, given that all strains resistant to ciprofloxacin—except for strain K-4—and ertapenem belonged to ST131. Interestingly, our results show that ST131 might also be relevant in eye infections as there are no known reports of the prevalence of eye infections caused by *E. coli* ST131 strains.

By comparing the virulence profiles with phylogenetic analysis, we observed that strains from groups B2 and F showed greater gene diversity and a higher number of genes per strain than those observed in other groups, thus reinforcing the notion that strains of group B2 usually carry a higher number of virulence genes, especially those associated with extraintestinal infections.

Studies assessing the distribution of virulence factors among ExPEC strains indicate a great diversity of genes with variable frequencies [12,44,45]. Due to the lack of studies regarding ocular *E. coli* virulence, we cannot draw a robust comparison of virulence profiles of ocular *E. coli* collections. However, by comparing our results to other published virulence profiles of ExPEC strains, we can observe some peculiar findings, such as the high prevalence of siderophore-encoding genes among our strains, such as *iucD, irp2,* and especially, *sitA* [46,47].

The *sitA* gene encodes an iron-manganese transporter that, according to some authors, is relevant to the pathogenicity of avian pathogenic *E. coli* [APEC] [48] and NMEC [49] strains. Its main function is to enable iron uptake and is especially important to bacterial survival in iron-deficient sites. While present in many ExPEC strains, a high prevalence such as the one presented is not usual. Regarding *E. coli* eye pathogenicity, this siderophore might play an important role in establishing an infection, given that the human eye surface has a very low amount of iron available and most of it is associated with lactoferrin as a measure to inhibit microbial growth [50]. These siderophores and iron-uptake systems might benefit bacterial growth by capturing all iron available on the lacrimal film and thus allowing the bacteria to infect the eye surface successfully.

Adhesins are relevant virulence factors to the pathogenesis of *E. coli* infections, given that they enable adhesion between bacteria and host cells. Many known genes carried by *E. coli* strains encode either fimbrial or afimbrial adhesins, such as *pap, sfa, fim*, *yfcV,* and *afa* [51,52,53]. Our strains were positive for many of these genes, with *ecp*, which encodes a subunit of the *E. coli* common pilus [54], and *fimA*, which encodes a subunit of the type I fimbria [55], being the first and second most prevalent among our strains, respectively. The high prevalence of *fimA* is relevant given the role of type I fimbria in *E. coli* extraintestinal diseases pathogenesis– by adhering to various types of epithelial cells, favoring biofilm-formation [56,57,58]—and evading extracellular antibiotics [59]. While it is not understood how it could affect *E. coli* ocular pathogenesis, it is known to confer to *Serratia marcescens* strains, an emerging ocular pathogen of the order Enterobacterales, the ability to adhere to HCEC in vitro [60]. Therefore, it is possible to presume that the type I fimbria might also benefit *E. coli* virulence in the human eye. However, it is worth mentioning that the presence of genes that are part of the *fim* operon is not always an indicator of its production. A simple agglutination test performed with our strains indicated that, despite the high prevalence of *fimA*, only nine strains—namely strains K-1, K-3, K-5, K-6, C-4, C-5, C-6, C-9, and C-10—produced the fimbria [data not shown], strongly suggesting that other adhesins might mediate adhesion to the cornea.

Lastly, *ibeA* is a virulence gene associated with some ExPEC pathotypes, such as NMEC, APEC, and AIEC [61,62]. It encodes an invasin that has been linked to many functions, such as invasion of brain endothelial cells [61], intramacrophage survival [62], and H_2_O_2_ stress survival [63]. Regarding pathogenicity in the human host, it is suggested to play a role in the pathogenesis of neonatal meningitis by mediating traversal of the blood-brain barrier and giving access to the bloodstream [64]. Based on our results, we cannot conclude whether the presence of *ibeA* affects bacterial pathogenicity in the eye. However, given its association with more severe infections, the presence of *ibeA* among our strains is nevertheless relevant, despite its low prevalence.

*E. coli* α-hemolysin (HlyA) is a cytotoxic toxin capable of lysing several cells—namely erythrocytes, leukocytes, and renal tubular cells—by forming pores in the target cell membrane [65]. It is encoded by a four-gene operon—*hlyCABD*—and, although the intracellular pathways of HlyA action are not entirely known for all diseases, its expression has been linked to more severe extraintestinal infections, especially UTIs [66]. Though the presence of the *hlyA* gene is a relevant predictor for the hemolytic phenotype, a strain that carries *hlyA* may be non-hemolytic, like the strains C-3a, C-3b, and C-4 in our study. It has been reported that defects in the *hlyCBD* operon or *rfaH*, a gene responsible for encoding a transcriptional activator of hemolysin synthesis, despite the presence of the *hlyA* gene, lead to the absence of a hemolytic phenotype [67]. The α-hemolysis phenotype presented by strain K-3 is worth noting because it lacks the *hlyA* gene, which probably indicates that the presented phenotype might be related to mechanisms other than HlyA production. It has been proposed that cytolysin A, a silent hemolysin encoded by the *sheA/hlyE* gene and found in both non-pathogenic and pathogenic *E. coli* strains, could be responsible for a hemolytic phenotype in strains that lack other *E. coli* hemolysin genes, such as genes from the *hlyCABD* operon, even though its role in pathogenesis is also yet to be determined [68]. Furthermore, enterohemolysin is another common *E. coli* hemolysin, with its name deriving from being first described in strains isolated from the fecal matter of infants with gastroenteritis [69]. Enterohemolysin is commonly associated with EHEC and STEC strains, but it can also be produced by non-EHEC/STEC strains [70]. It can be encoded by the plasmid-borne operon *ehxCABD* or phage-associated *ehlyA* gene and is phenotypically different from HlyA—it forms smaller halos that only appear after overnight incubation, in contrast with the 3 h incubation period required for HlyA [71]. Despite this phenotypical differentiation, strain C-15 could not be considered enterohemolysin positive as it lacks the enterohemolysin-encoding gene.

It is currently unknown how *E. coli* hemolysins would affect the eye during an infection. In some cases, the role of hemolysins in the pathogenesis of eye infections is quite well described. Studies about how hemolysins produced by other species—such as *Staphylococcus aureus, Bacillus cereus, Pseudomonas aeruginosa,* and *Enterococcus faecalis*– affect the eye do indicate that compared to non-hemolytic strains, a more aggressive infection takes place, regardless of the infected eye structure [72,73,74,75,76]. A study about the *S. aureus* α-hemolysin role in infectious keratitis pathogenesis observed that it prevents corneal re-epithelialization, thus preventing the healing of corneal ulcers, and is essential for intracellular bacterial invasion, even though it is not solely responsible for the invasion [72]. In another study, increased corneal opacity and leukocytic invasion of the corneal stroma were observed in vivo by injecting purified *P. aeruginosa* hemolysin intracorneally in rabbits [73]. In endophthalmitis models, studies about hemolytic *B. cereus* and *E. faecalis* strains indicated increased retinal damage and local immune response in eyes infected with hemolytic *E. faecalis* [74]. In contrast, hemolytic *B. cereus* caused severe retinal damage slightly faster than non-hemolytic strains [75,76]. Given those examples, it is possible to assume that, while not essential, hemolysins increase bacterial virulence in the eye and help the bacteria to thrive more quickly on the site. Furthermore, given our results, it is possible to assume that *E. coli* does not depend on hemolysis to infect and survive on the ocular surface, even though more studies are needed to understand if and how *E. coli* hemolysins affect the pathogenesis of eye infections.

In the literature, the susceptibility of *E. coli* isolates recovered from ocular infections to ciprofloxacin is reportedly variable. For example, in the study developed by Mohammed et al. [77], all isolates were sensitive to ciprofloxacin, whereas the studies by Getahun et al. [78] and Ranjith et al. [79] demonstrated resistance rates of 16.7% and 58%, respectively. In recent years, an increase in the number of *E. coli* strains resistant to ciprofloxacin has been observed. According to studies, this increase is mainly due to the dissemination of international clones ST131 (Subclones 131-H30-R and 131-H30Rx) and ST1193, which have mutations in the *gyrA* and *parC* genes that confer resistance to fluoroquinolones [80].

In contrast, other studies observed resistance phenotypes to other antimicrobials, unlike our isolates, which were multi-susceptible [77,78,79,81,82]. Worryingly, some reports have already reported multi-drug-resistant (MDR) *E. coli* strains mainly as a result of extended-spectrum B-lactamase (ESBL) production [25,83], which consequently limits the therapeutic options for treatment. However, it is worth noting that most eye infections are treated using topical antibiotics that, in turn, offer a localized treatment with a higher antibiotic concentration than that achieved during systemic treatment. Furthermore, it has been proposed that resistance observed with systemic breakpoints may overestimate the real potential ocular resistance [84]. Therefore, while still relevant, our results cannot indicate a possible treatment failure, as all breakpoints used are based on the antibiotic concentration found during systemic treatment.

Bacterial adherence to host cells is an essential part of pathogenesis, as it allows the bacteria to successfully colonize the infection site, despite the action of external factors. It can be mediated by many factors, such as adhesive fibers named “fimbriae” that facilitate adhesion to host cells and interbacterial aggregation [54,58,85]. Adhesion is highly relevant regarding eye infections, as the eyelids’ constant motion prevents non-adherent bacteria from colonizing the eye surface.

As previously mentioned, type I fimbria is pivotal for other species to adhere to corneal cells. In our results, most of the nine strains that produced type I fimbria presented a slightly higher adherence than strains that do not produce this fimbria. Furthermore, strain C-6 appears to produce type I fimbria and presented a higher level of adherence to corneal cells and interbacterial aggregation. All these findings might be related to the production of type I fimbria, but more studies are required to confirm this. Furthermore, our results also point to the involvement of other *E. coli* adhesins, with strain C-15 being an example of this. Although this strain lacks the *fimA* gene, it still presented a high level of adherence. Other adhesin-encoding genes carried by this strain, such as *afaBCIII* and *yfcV,* might influence its adherence to HCECs.

While the interaction between *E. coli* strains and human corneal cells is vastly unknown, our results indicate that ExPEC strains can adhere to corneal cells and more than one type of adhesins might mediate such adherence. Although more studies are still required to understand how *E. coli* adhesins work on the ocular surface, our results provide some insight into the ocular pathogenesis of *E. coli*.

Clinically, the uropathogenic potential of more than half of our strains might be the most relevant finding, as it might be an indicator of a possible link between UTI and eye infections. We cannot draw any conclusions regarding these findings as we did not have access to the patients’ clinical history. Therefore, we cannot know if they developed an eye infection during or shortly after a UTI.

The main limitation we faced was the scarcity of published works about ocular *E. coli*, as it hinders the possibility of comparing our results to those obtained from a strain of the same niche. This might make it difficult to identify patterns among ocular *E. coli* strains that could help us understand if there are specific characteristics shared between them. Secondly, while in vitro results are essential, there is an important gap between how a bacterial strain behaves in vitro and how it would behave in vivo, especially concerning the eye. Numerous factors that could prevent bacterial colonization are absent in an in vitro assay. Therefore, while our results point to some factors that could help bacterial colonization of the eye surface, it is not possible to firmly conclude about *E. coli* eye pathogenicity based solely on molecular and in vitro results.

## 5. Conclusions

Our findings indicate that *E. coli* strains isolated from eye infections display an extensive genetic variety and a considerable diversity regarding their virulence profile. In addition, the absence of strains from other pathotypes and hybrid strains might indicate that to thrive on the human eye, ExPEC virulence mechanisms, while not intrinsically required, might play an important role. Furthermore, our findings present an interesting insight into how *E. coli* might infect the human eye. Still, more studies are required to understand better how *E. coli* reaches the human eye to help prevent future infections.

## Figures and Tables

**Figure 1 microorganisms-10-01084-f001:**
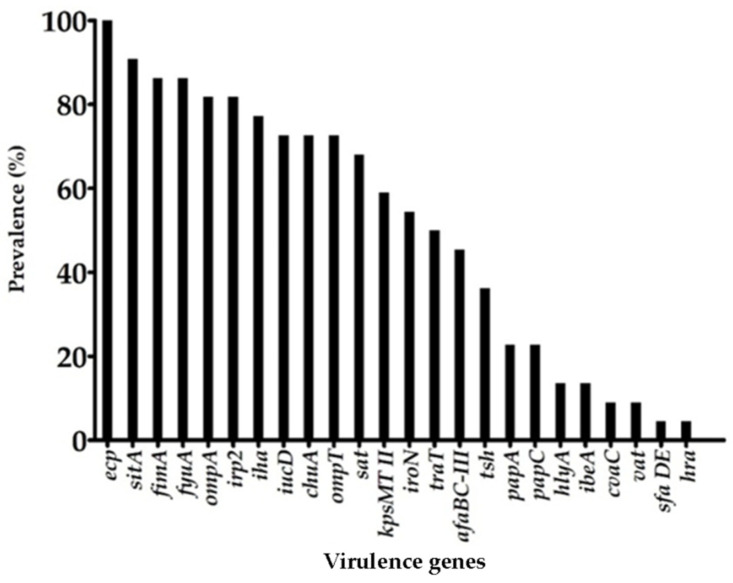
Virulence genes prevalence among *Escherichia coli* strains isolated from eye infections. The *aggR, afaE-VIII, bfpB, bmaE, cf29A, cnf1, eae, eltA, escV, estA, invE, ireA, kpsMTIII, stx-1,* and *stx-2* genes were absent in all strains.

**Figure 2 microorganisms-10-01084-f002:**
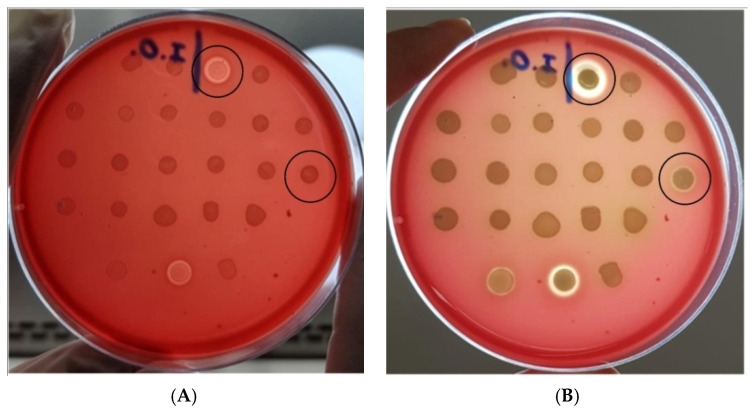
Blood agar plate inoculated with *Escherichia coli* strains isolated from eye infections (**A**) after 3 h of incubation; (**B**) after 24 h of incubation. Positive strains are circled in black. Controls at bottom row (from left to right): Shiga-toxin producing *E. coli* (STEC) strain EDL933 (enterohemolysin producer), UPEC strain CFT073 (alpha-hemolysin producer), and *Klebsiella* spp. K33 (non-hemolytic strain). The ocular *E. coli* strain C-3a was not included in this test.

**Figure 3 microorganisms-10-01084-f003:**
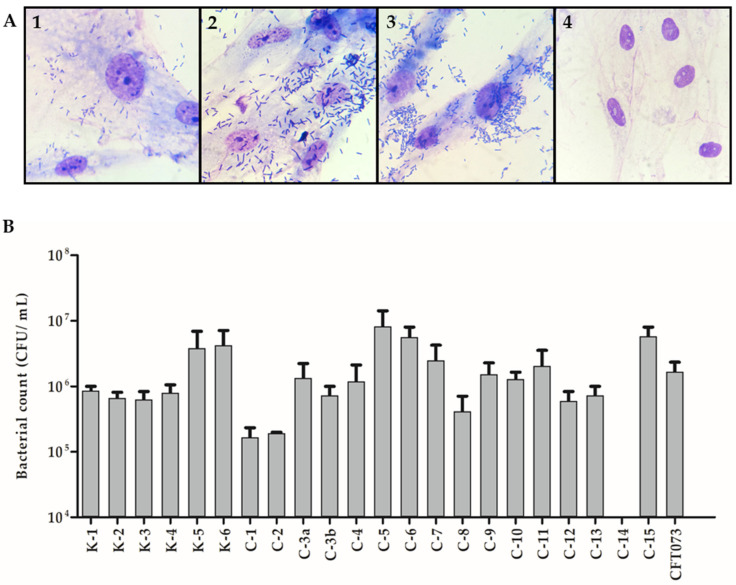
Adherence of *Escherichia coli* strains isolated from eye infections to human corneal epithelial cells (HCECs). (**A**) Representative examples of different levels of *E. coli* strains adherence to HCEC under oil immersion light microscopy: (1) strain C-8, low adherence; (2) strain C-15, high adherence; (3) strain C-6, high adherence and aggregative adherence pattern; (4) non-infected human corneal epithelial cells. (**B**) Bacterial adherence based on quantitative assay results. The prototype strain CFT073 was used as a control for ExPEC adherence.

**Table 1 microorganisms-10-01084-t001:** Phylogenetic profile and sequence type of *Escherichia coli* strains isolated from eye infections.

Strain	Disease	Phylogroup	Sequence Type ^a^
K-1	Keratitis	A	-
K-2	Keratitis	B1	-
K-3	Keratitis	B2	-
K-4	Keratitis	B2	-
K-5	Keratitis	B2	ST131
K-6	Keratitis	C	-
C-1	Conjunctivitis	A	-
C-2	Conjunctivitis	A	-
C-3a	Conjunctivitis	B2	ST131
C-3b	Conjunctivitis	B2	ST131
C-4	Conjunctivitis	B2	ST131
C-5	Conjunctivitis	B2	ST131
C-6	Conjunctivitis	B2	ST131
C-7	Conjunctivitis	B2	ST131
C-8	Conjunctivitis	B2	ST131
C-9	Conjunctivitis	B2	ST131
C-10	Conjunctivitis	B2	ST131
C-11	Conjunctivitis	B2	ST131
C-12	Conjunctivitis	D	-
C-13	Conjunctivitis	D	-
C-14	Conjunctivitis	F	-
C-15	Conjunctivitis	F	-

^a^, A negative result for all tested sequence types was represented by “-”.

**Table 2 microorganisms-10-01084-t002:** Virulence profiles of *Escherichia coli* strains isolated from eye infections.

Strain	Phylogroup	Virulence Markers	Intrinsic Virulence ^a^	Uropathogenicity ^a^	Predicted Pathotype
K-1	A	*ecp, sitA, fimA, ompA, ompT, cvaC*	*iucD, kpsMTII, papA*	*-*	ExPEC
K-2	B1	*ecp, fimA, ompA, ompT*	*-*	*-*	-
K-3	B2	*ecp, fimA, ompA, irp2, iha, ompT, sat, traT, hra*	*kpsMTII, papA, papC, sfaDE*	*fyuA, chuA, yfcV, vat*	ExPEC/UPEC
K-4	B2	*ecp, sitA, fimA, ompA, irp2, iha, ompT, sat, tsh*	*iucD, kpsMTII*	*fyuA, chuA, yfcV, vat*	ExPEC/UPEC
K-5	B2	*ecp, sitA, fimA, ompA, irp2, iha, ompT, sat, iroN, traT, tsh*	*iucD, papC*	*fyuA, chuA*	ExPEC
K-6	C	*ecp, sitA, fimA, ompA, irp2, ompT, cvaC*	*papA*	*fyuA, yfcV*	-
C-1	A	*ecp, sitA, fimA, irp2, iha, sat, iroN, traT, tsh*	*iucD, afaBCIII*	*fyuA*	ExPEC
C-2	A	*ecp, sitA, ompA, irp2, iha*	*iucD, afaBCIII*	*fyuA*	ExPEC
C-3a	B2	*ecp, sitA, fimA, ompA, irp2, iha, sat, iroN, ibeA, hlyA*	*afaBCIII*	*fyuA, chuA, yfcV*	UPEC
C-3b	B2	*ecp, sitA, fimA, ompA, iha, ompT, sat, iroN, hlyA*	*kpsMTII, afaBCIII*	*fyuA, chuA*	ExPEC
C-4	B2	*ecp, sitA, fimA, ompA, iha, ompT, sat, iroN, traT, ibeA, hlyA*	*iucD, kpsMTII, afaBC III, papC*	*fyuA, chuA, yfcV*	ExPEC/UPEC
C-5	B2	*ecp, sitA, fimA, ompA, irp2, iha, ompT, sat, iroN, traT*	*iucD*	*chuA, yfcV*	-
C-6	B2	*ecp, sitA, fimA, irp2, iha, ompT, tsh*	*iucD, kpsMTII*	*fyuA, chuA, yfcV*	ExPEC/UPEC
C-7	B2	*ecp, sitA, fimA, ompA, irp2, iha, ompT, sat, iroN, traT, tsh*	*iucD, kpsMTII, afaBCIII*	*fyuA, chuA, yfcV*	ExPEC/UPEC
C-8	B2	*ecp, sitA, fimA, ompA, irp2, iha, ompT, sat*	*iucD, kpsMTII*	*fyuA, chuA, yfcV*	ExPEC/UPEC
C-9	B2	*ecp, sitA, fimA, ompA, irp2, iha, sat, tsh*	*iucD*	*fyuA, chuA, yfcV*	UPEC
C-10	B2	*ecp, sitA, fimA, ompA, irp2, iha, ompT, sat, iroN, traT, tsh*	*iucD, kpsMTII, afaBCIII*	*fyuA, chuA, yfcV*	ExPEC/UPEC
C-11	B2	*ecp, sitA, fimA, irp2, iha, ompT, sat, iroN, traT, tsh, ibeA*	*iucD, kpsMTII*	*fyuA, chuA, yfcV*	ExPEC/UPEC
C-12	D	*ecp, sitA, ompA, irp2, iroN, traT*	*iucD, kpsMTII, afaBCIII*	*fyuA*	ExPEC
C-13	D	*ecp, sitA, fimA, irp2, iha, iroN, traT*	*iucD, kpsMTII, afaBCIII*	*fyuA, chuA*	ExPEC
C-14	F	*ecp, sitA, fimA, ompA, irp2, iha, ompT, sat*	*iucD, kpsMTII, papA, papC*	*fyuA, chuA, yfcV*	ExPEC/UPEC
C-15	F	*ecp, sitA, ompA, irp2, ompT, sat, iroN, traT*	*kpsMTII, afaBCIII, papA, papC*	*fyuA, chuA, yfcV*	ExPEC/UPEC

^a^, Genes associated with intrinsic virulence and uropathogenicity were defined by Johnson et al. [10] and Spurbeck et al. [11], respectively. Therefore, pathotype predictions for ExPEC and UPEC were based on molecular analyses.

## Data Availability

Not applicable.

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
