# Peer review of "Evaluation of the Pathogenic Potential of Escherichia coli Strains Isolated from Eye Infections"

_microorganisms, 2022, doi:10.3390/microorganisms10061084_

Round 1

Reviewer 1 Report

Although the topic discussed in this paper entitled:" Evaluation of the Pathogenic Potential of Escherichia coli Strains Isolated From Eye Infections" is of interest to readers, and scientifically relevant, some minor revisions are needed before publication.

Revisions below

Abstract

"polymerase chain reactions" insert acronym

"sequence typing" initial capital letters

"corneal cells" indicate cell line

Introduction

ExPEC  checks whether the acronym has been made explicit at least once at the beginning of the text

Materials and Methods

2.4 Virulence Genes profiles

Nascimento et al. [18] work please delete the word “work”

2.7. Cell Culture and In-vitro Adherence to Human Corneal Epithelial Cells Assay

For quantitative analysis, assays were performed following the methodology…. Replace “methodology” with protocol

3.2. Virulence Genes Profile

From: Based on the proposed method of Johnson et al. [8]……. To ….. lacking all the markers above. Hard to read sentence please rewrite

Discussion

From: Ocular diseases due to E. coli continue to be understudied….. To: …. extraintestinal pathogenicity that these strains bear. Hard to read paragraph please rewrite

Caption Table 3: enter the meaning of S and R used in the table

Author Response

Point 1: Abstract; "polymerase chain reactions" insert acronym; "sequence typing" initial capital letters; "corneal cells" indicate cell line

Response 1: All corrections were performed in the indicated places.

Point 2: Introduction: ExPEC  checks whether the acronym has been made explicit at least once at the beginning of the text.

Response 2: We have inserted the meaning of the acronym at the beginning of the text.

Point 3. Materials and Methods:

2.4 Virulence Genes profiles

Nascimento et al. [18] work please delete the word “work”

Response 3: The word was deleted.

Point 4: Materials and Methods:

2.7. Cell Culture and In-vitro Adherence to Human Corneal Epithelial Cells Assay

For quantitative analysis, assays were performed following the methodology…. Replace “methodology” with protocol

Response 4: Replaced, as suggested.

Point 5:  3.2. Virulence Genes Profile

From: Based on the proposed method of Johnson et al. [8]……. To ….. lacking all the markers above. Hard to read sentence please rewrite.

Response 5: The sentence was rewritten as follows:

“Seventeen strains presented at least two of the five intrinsic virulence markers proposed by Johnson et al. [8] to define ExPEC intrinsic virulence. Of the remaining five strains, four carried one of these markers, while strain K-2 was devoid of all five markers.”

Point 6: Discussion

From: Ocular diseases due to E. coli continue to be understudied….. To: …. extraintestinal pathogenicity that these strains bear. Hard to read paragraph please rewrite.

Response 6: The paragraph was rewritten as follows:

“The E. coli strains associated with eye infections are understudied, despite their continuous – yet not so frequent – isolation from this site. To enhance the knowledge regarding this topic, in the present work, we performed a molecular and phenotypic evaluation of diverse virulence aspects in a collection of E. coli strains isolated from cases of keratitis and conjunctivitis.

It is a consensus that E. coli strains isolated from extraintestinal infections share diverse traits related to their capacity to cause infections out of the intestinal site. In this sense, phylogroup B2 is pointed out as the most frequent group associated with extraintestinal infections due to the higher prevalence of specific virulence factors related to the extraintestinal pathogenicity that these strains commonly bear.”

Point 7: Caption Table 3: enter the meaning of S and R used in the table

Response 7: The meaning was inserted in Table 3´s (now Table S2) caption.

Reviewer 2 Report

The above noted manuscript describes extensive investigations with ocular gram negative organisms - i.e., E. coli related to pathogenic potential.  The manuscript is well done and presented and I do not have any major concerns.  the following are a few suggestions for consideration by the authors:

1) introduction...Enterobacteriaceae now belong to Enterobacteriales

2) the 2nd last paragraph in the introduction on "hybrids" may be better placed in the discussion

3) under antibiotic susceptibility testing, a reference or website to the BrCAST should be provided as these criteria would be less familiar to most readers when compared to CLSI or EUCAST

4) i fully realize susceptibility testing breakpoints do not take into account topical antibiotic administration as in ophthalmology.  it might be worth a comment on this in the paper that "resistance" may not necessarily mean clinical failure due to the high drug concentrations delivered during topical therapy

5) Table 3  My preference is always for MIC values rather than just a designation of S or R.  unfortunately, MIC values were not determined for these strains but rather zone sizes.  Given that with the exception of ciprofloxacin the organisms were broadly susceptible to the drugs tested (one resistant to gent and one resistant to erta), this table does not add to the manuscript and should be deleted.  It is sufficient to summarize the susceptibility data in the text.

6) a statement on the clinical implications of the findings reported would be useful

7) a section on study limitations should be included.

A very good paper representing a substantial amount of work.  Congratulations.

Author Response

Point 1. introduction...Enterobacteriaceae now belong to Enterobacteriales

Response 1: As suggested, we inserted, in this phrase, the “Enterobacterales order” to which the Enterobacteriaceae family belongs.

Point 2. the 2nd last paragraph in the introduction on "hybrids" may be better placed in the discussion

Response 2: We prefer to keep the information on “hybrid strains” in the introduction because we looked for this type of strain in the study. Since we did not find any “hybrids” among the strains analyzed, they were not further discussed in the manuscript.

Point 3. under antibiotic susceptibility testing, a reference or website to the BrCAST should be provided as these criteria would be less familiar to most readers when compared to CLSI or EUCAST

Response 3: We modified the text to make this information clearer. BrCAST is the authorized translation of EUCAST guidelines and breakpoints. Therefore, we introduced the EUCAST website address as a reference for this topic.

Point 4. i fully realize susceptibility testing breakpoints do not take into account topical antibiotic administration as in ophthalmology.  it might be worth a comment on this in the paper that "resistance" may not necessarily mean clinical failure due to the high drug concentrations delivered during topical therapy.

Response 4: As suggested, we have inserted a comment in the discussion as follows:

“However, it is worth noting that most eye infections are treated using topical antibiotics that, in turn, offer a localized treatment with a higher antibiotic concentration than that achieved during systemic treatment. Furthermore, it has been proposed that resistance observed with systemic breakpoints may overestimate the real potential ocular resistance [67]. Therefore, while still relevant, our results cannot indicate a possible treatment failure, as all breakpoints used are based on the antibiotic concentration found during systemic treatment.”

Point 5. Table 3 My preference is always for MIC values rather than just a designation of S or R.  unfortunately, MIC values were not determined for these strains but rather zone sizes.  Given that with the exception of ciprofloxacin the organisms were broadly susceptible to the drugs tested (one resistant to gent and one resistant to erta), this table does not add to the manuscript and should be deleted.  It is sufficient to summarize the susceptibility data in the text.

Response 5: We agree with your comment. We moved Table 3 to the Supplementary material as Table S2.

Point 6. a statement on the clinical implications of the findings reported would be useful.

Response 6: As suggested, we introduced a statement as follows (page 16):

“Clinically, the uropathogenic potential of more than half of our strains might the most relevant finding, as it might be an indicator of a possible link between UTI and eye infections. We cannot draw any conclusions regarding these findings as we did not have access to the patients’ clinical history. Therefore, we cannot know if they developed an eye infection during or shortly after a UTI.”

Point 7. a section on study limitations should be included.

Response 7: As suggested, we introduced a section as follows (page 16):

“The main limitation we faced was the scarcity of published works about ocular E. coli, as it hinders the possibility of comparing our results to those obtained from a strain of the same niche. This might make it difficult to identify patterns among ocular E. coli strains that could help us understand if there are specific characteristics shared between them. Secondly, while in vitro results are essential, there is an important gap between how a bacterial strain behaves in vitro and how it would behave in vivo, especially concerning the eye. Numerous factors that could prevent bacterial colonization are absent in an in vitro assay. Therefore, while our results point to some factors that could help bacterial colonization of the eye surface, it is not possible to firmly conclude about E. coli eye pathogenicity based solely on molecular and in vitro results.”

A very good paper representing a substantial amount of work.  Congratulations.

Thank you so much for your support and useful suggestions.